# ANORAND: DEEP LEARNING-BASED SEMI-SUPERVISED ANOMALY DETECTION WITH SYNTHETIC LABELS

## ABSTRACT

Anomaly detection, or more generally outlier detection, is one of the most popular and challenging topics in theoretical and applied machine learning. The main challenge is that in general we have access to very few labeled data or no labels at all. In this paper, we present a new semi-supervised anomaly detection method called **AnoRand** by combining a deep learning architecture with random synthetic label generation. The proposed architecture has two building blocks: (1) a noise detection (ND) block composed of feed forward perceptron and (2) an autoencoder (AE) block. The main idea of this new architecture is to learn one class (e.g. the majority class in case of anomaly detection) as well as possible by taking advantage of the ability of auto encoders to represent data in a latent space and the ability of Feed Forward Perceptron (FFP) to learn one class when the data is highly imbalanced. First, we create synthetic anomalies by randomly disturbing a few samples (e.g., 2%) from the training set. Second, we use the normal and synthetic samples as input to our model. We compared the performance of the proposed method to 17 state-of-the-art unsupervised anomaly detection method on synthetic datasets and 57 real-world datasets. Our results show that this new method generally outperforms most of the state-of-the-art methods and has the best performance (AUC ROC and AUC PR) on the vast majority of reference datasets.

## 1 INTRODUCTION

Anomaly detection is one of the most exciting and challenging subject in theoretical and applied machine learning. nowadays, deep learning methods are increasingly used for anomaly detection. One of the main challenges in anomaly detection or more generally outlier detection is that you don't have enough samples labeled as anomalous. In this situation, most of the classical machine learning methods fail to learn the minority (anomaly) class, which is most of the time the class of interest. Furthermore, label accuracy can be problematic, with some anomalies mislabeled as normal and vice versa. Unsupervised methods have gained traction because they don't require labeled data. They model the normal sample distribution and detect anomalies by identifying outliers. However, unsupervised methods have limitations; they assume anomalies are in low-density regions, relying on these assumptions for performance. Moreover, it has been shown Collin & De Vleeschouwer (2021); You et al. (2022) that the reconstruction-based anomaly detection method such as auto encoders leads to a high number of false alarms.

To detect anomalies and outliers in a semi-supervised way, we propose a method that combines a deep auto encoder and feed forward perceptrons (FFP). This new method that we call **AnoRand**, jointly optimizes the deep autoencoder and the FFP model in an end-to-end neural network fashion. AnoRand has two building blocks: a Feed Forward Perceptron block and an autoencoder block. The inspiration for this method comes from two ideas: (i) Anomaly detection problem can be cast as a supervised learning between normal samples and noise Steinwart et al. (2005); (ii) Supervised algorithms tend to learn only the majority class in presence of highly imbalance data Anand et al. (1993). The main objective of thi new method is to learn one class (e.g. the majority class in case of anomaly detection) as well as possible by taking advantage of the ability of auto encoders to represent data in a latent space and the ability of Feed Forward Perceptron (FFP) to learn one

class when the data is highly imbalanced. In our method, the FFP block has a role in informing and strengthening the capacity of the auto-encoder block to embed the normal samples. We compared the performance of the proposed method to 17 state-of-the-art unsupervised anomaly detection method on synthetic data sets and 57 real-world data sets from the ADBench benchmark paper Han et al. (2022). Our results show that this new method generally outperforms most of the state-of-the-art methods and has the best performance (AUC ROC and AUC PR) on the vast majority of reference datasets. In particular, AnoRand outperforms Deep auto encoder, Variational auto encoder and MLP even though they have the same kind of building blocks.

The main contributions of this paper are:

- Novel Anomaly Detection Method (AnoRand): This new method does not require any assumptions about the underlying shape of the decision boundary that separates normal data points from anomalous ones. This flexibility makes it highly adaptable to various real-world datasets and scenarios.

- Learning from Limited Information: AnoRand learns a reliable decision boundary using only normal samples and noisy version of few of them. This feature is particularly valuable in situations where obtaining labeled anomalous data is challenging or costly.

- Extensive Benchmarking and Superior Performance: To assess the effectiveness of AnoRand, we conducted comprehensive experiments on 57 diverse real-world anomaly detection benchmark datasets. The results demonstrate that AnoRand achieves state-of-the-art performance on the majority of these datasets. Its robustness and versatility make it a promising choice for a wide range of applications.

## 2 RELATED WORKS

**Anomaly detection.** Anomaly detection, also called outlier detection, is a technique in machine learning and statistics for identifying unusual data points. These outliers are instances that significantly deviate from the expected normal behavior found in most data. The main goal is to detect and investigate these rare events, which may indicate potential issues or unusual patterns in the data. It finds applications in various industries, including healthcare, finance, manufacturing, cybersecurity, and more. Anomalies can result from equipment malfunctions, data errors, fraud, or unusual behavior. In the literature, there are three main types of anomaly detection methods: supervised, semi-supervised, and unsupervised.

**Fully supervised and Semi-supervised algorithms.** These methods use labeled or partially labeled data to train a classifier capable of distinguishing normal from anomalous samples. Fully supervised methods require complete access to labeled data and excel when high-quality labeled data and a clear distinction between normal and anomalous samples are available. In contrast, semi-supervised methods offer more flexibility and adaptability in situations with limited labeled data or when detecting novel anomalies is required. They can outperform supervised methods as they leverage unlabeled data to learn the normal data distribution and are more robust to noise. However, both fully supervised and semi-supervised methods are constrained by the quality and representativeness of labeled data. They may struggle when encountering novel anomalies not seen during training and can be challenged by imbalanced class distributions, leading to a high rate of false negatives. Imbalanced classes can introduce bias, causing algorithms to fail in learning the minority class effectively. These algorithms include classical methods such as SVM, random forest Zhao & Hryniewicki (2018); Bayes (1763); Cortes & Vapnik (1995); Breiman (2001); Chen & Guestrin (2016); Ke et al. (2017); Prokhorenkova et al. (2018) and deep learning based methods Akcay et al. (2019); Ruff et al. (2020); Pang et al. (2018; 2019a); Zhou et al. (2021); Rosenblatt (1958); Gorishniy et al. (2021); Pang et al. (2019b).

**Unsupervised algorithms.** Unsupervised methods on the other hand are more flexible, especially when labeled anomalies are scarce or when novel anomalies need to be detected. They have gained popularity due to the fact that they don't required labeled data. They make the assumption that anomalies are located in low-density regions, and therefore their performance is highly dependent on the alignment of these assumptions and the underlying anomaly type. They are effective when the input data and the algorithm assumption(s) meet. The main limitations of unsupervised methods is that, the detection accuracy may be lower than supervised or semi-supervised methods, especially in cases with high noise or complex data distributions. They also struggle with class imbalance,

as it treats all anomalies equally. These algorithms can be grouped into two categories: classical algorithms and deep learning-based algorithms. Classical algorithms include distribution and distance-based unsupervised methods Ramaswamy et al. (2000); Liu et al. (2008); Schölkopf et al. (1999); Li et al. (2022); Breunig et al. (2000); He et al. (2003); Tang et al. (2002); Goldstein & Dengel (2012); Kriegel et al. (2009); Li et al. (2020); Shyu et al. (2003); Pevnỳ (2016); Han et al. (2022); Zhao et al. (2019); Steinwart et al. (2005). Deep learning based algorithms includes algorithm that use deep learning representation to cluster data into homogeneous classes. Deep Support Vector Data Description (DeepSVDD) Ruff et al. (2018) uses the idea of OCSVM by training a neural network to learn a transformation that minimizes the volume of a hypersphere in the output space that encloses the samples of one class. All the samples that are far from the center of the hypersphere are labeled as anomalies. Deep Autoencoding Gaussian Mixture Model (DAGMM) Zong et al. (2018) optimizes jointly a deep autoencoder and a Gaussian mixture model in the same learning loop. Auto encoders have also been used for anomaly detection by some authors Kingma & Welling (2013); Zhou et al. (2020); Zavrtanik et al. (2021); Shi et al. (2021). They used the reconstruction error as an anomaly score. It has been shown Collin & De Vleeschouwer (2021); You et al. (2022) that reconstruction-based anomaly detection methods such as auto encoders lead to high number of false alarms. Indeed, auto encoder tend to produced blurry output. This behavior can lead to a kind of smoothing or blurring the anomalies such that it will look like normal samples.

## 3    PROBLEM STATEMENT

Most anomaly detection methods in the literature face challenges such as algorithm selection, parameter tuning, heavy reliance on labels, and unsatisfactory performance. Label-based approaches depend on the quality and representativeness of labeled data, struggling with novel anomalies and class imbalances. Unsupervised methods, though label-independent, often underperform compared to supervised or semi-supervised methods, particularly with noisy or complex data. Deep learning-based unsupervised methods Collin & De Vleeschouwer (2021); You et al. (2022); Xu et al. (2018), like autoencoders, can yield high false alarms due to "blurry" reconstructed outputs, making subtle anomaly detection difficult.

In this study, our objective is to overcome certain limitations observed in previous approaches, such as label dependency and the need for predefined assumptions. We introduce a novel semi-supervised anomaly detection method named Anorand to address these challenges. Our method operates by harmoniously optimizing a deep autoencoder (AE) and a Feed Forward Perceptrons (FFP) within a single neural network architecture. This method is closely related to the theoretical work of Steinwart et al. Steinwart et al. (2005), where they have shown that an anomaly detection problem can be cast as supervised learning between normal samples and noise. Unlike their work, we have learned the discriminative function in a deep learning framework.

## 4    ANORAND METHOD AND IMPLEMENTATION DETAILS

### 4.1    PROPOSED ARCHITECTURE

Lets first define $(x, y) = \{(x_1, y_1), (x_2, y_2), \ldots (x_N, y_N)\}$ in the machine learning framework such that $y_i$ is the target (label) vector and $x_i \in R^d$ the feature vector for the $ith$ sample.

In this paper, we proposed to combine an autoencoder architecture with a fully connected architecture to detect anomalies. The proposed architecture has two building blocks: a noise detection (ND) block composed of Feed Forward Perceptron (FFP) and an autoencoder (AE) block. We call this new method **AnoRand**. AnoRand jointly optimizes a deep autoencoder and a FFP model in an end-to-end neural network fashion. lassesThe joint optimization in AnoRand empowers the autoencoder to escape from suboptimal local optima, ultimately reducing reconstruction errors. Autoencoders excel in their ability to learn compact data representations in a latent space. However, their effectiveness in anomaly detection can be limited in cases of imbalanced lasses. Autoencoders, by design, aim to capture essential features of the input data in the encoding while discarding some fine-grained details. This process often results in the reconstructed output being somewhat "blurry" compared to the original input. The blurring effect is particularly noticeable in image data, where autoencoders are widely applied. This blurriness can make it challenging to distinguish anomalies from normal

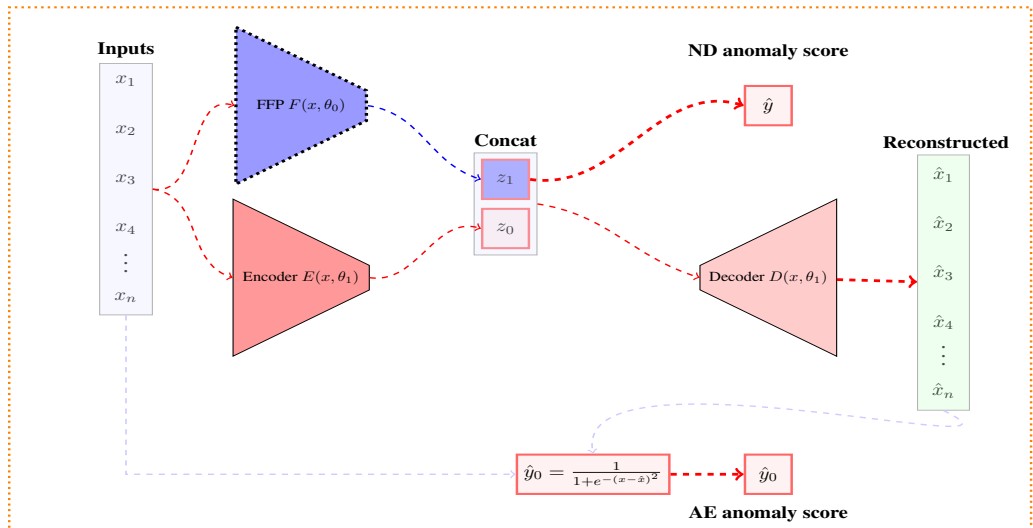

Figure 1: Overall architecture of AnoRand

data, especially if anomalies involve subtle deviations or fine details. Anomalies may appear less distinct from normal samples in the reconstructed data. This can lead to anomalies being incorrectly classified as normal because their characteristics are smoothed out during reconstruction.

AnoRand addresses this limitation by adding the FFP's last-layer output to the AE latent vector. Indeed, the FFP model is particularly adapted for supervised balanced data, but in the presence of skewed class distributions, FFP tend to learn the characteristics of the majority class. AnoRand's primary objective is to learn one class of data, typically the majority class in the context of anomaly detection, as effectively as possible. It leverages the strengths of autoencoders to represent data in a latent space and the capability of Feed Forward Perceptrons (FFP) to learn a single class when the data is highly imbalanced. AnoRand method is performed in two steps: (i) **Synthetic anomaly generation:** In the first step, synthetic anomalies are created as described in Section 4.2. This process enriches the training dataset with anomalous instances, effectively allowing the model to learn and adapt to a wider range of anomalies. (ii) **Model input and training:** In the second step, both normal data samples and synthetic anomalies are used as input to the AnoRand model. The model is then trained on this combined dataset to distinguish between normal data and anomalies. The FFP block, plays a pivotal role, serves as a critical component within the AnoRand architecture. We refer to this block as the "noise detection block" because it exhibits a remarkable ability to identify instances with anomalies, particularly when synthetic labels are generated with a high level of noise. Let's denote by $z_1 = F(x, \theta_0)$ the mapping of the input features by FFP and $z_0 = E(x, \theta_1)$ its mapping by the encoder from the auto encoder architecture. $\theta_1$ is the weights of the Feed Forward Perceptron (FFP) and $\theta_0$ the weights of the auto encoder block. The final latent vector of the autoencoder block is defined as follows:

$$z = (z_0, z_1) = (F(x, \theta_0), E(x, \theta_1)) \tag{1}$$

The last layer of the FFP, denoted as $z_1 = F(x, \theta_0)$, produces an output based on the features it has learned. The combination of $z_0$ and $z_1$ allows for the integration of information captured by the autoencoder (data-specific features) with the features learned by the FFP (task-specific or high-level features). This combined vector is particularly powerful when we want to perform tasks like generating data samples that are consistent with both the learned data distribution and the task-specific features. The final model has one input and two outputs. The model takes as input the features X and the synthetic labels Y generated as described in subsection 4.2. The first output $\hat{y}$ is the probability of the sample being a noisy or anomaly sample estimated by the noise detection block, the second output $\hat{x}$ is the reconstructed signal of the autoencoder block. The full network architecture is described in Figure 1.

## 4.2 SYNTHETIC LABEL GENERATION

To create our semi-supervised model, we introduced synthetic anomalies into our training dataset (see Appendix 8.1 for more details about synthetic anomaly generation). We began by randomly

selecting a subset of samples from the training data, denoted as $X_0$, with a subset $X_0^0$ representing anomalies. We then transformed $X_0^0$ into synthetic anomalies by adding noise to simulate real-world anomaly data. The noisy samples can be generated by using Gaussian noise or other sophisticated methods such as in Cai et al.Cai & Fan (2022). In our experiments, we used a hybrid approach that combined Gaussian noise with the Synthetic Minority Over-sampling Technique (SMOTE) Chawla et al. (2002) to generate synthetic anomalies. SMOTE oversampled $X_0^0$ until it constituted 5% of the total training dataset. This approach ensured that synthetic anomalies closely resembled the feature distribution of normal samples. The resulting training dataset, $X_{tr}$, consisted of three components: $X_1$ representing the majority of normal samples, $X_0^0$ as the foundation for synthetic anomalies, and $X_0^1$ representing the noisy synthetic anomalies created through SMOTE and Gaussian noise. This composition allowed $X_{tr}$ to contain 5% "anomalous" samples, including both originally selected $X_0^0$ samples and the synthetic anomalies created.

## 4.3 OBJECTIVE FUNCTION

The final loss function $\mathcal{L}(\theta)$ combines two individual loss components with a weight factor $w$.
$\mathcal{L}(\theta_0)$: This component represents the reconstruction error from the autoencoder (AE) block. It is computed for each data point $\mathbf{x}_i$ and its corresponding reconstructed output $\hat{x}_i$. This term encourages the AE to learn a meaningful representation of the normal samples.

$$\mathcal{L}(\theta_0) = \sum_{i=1}^{N} \mathcal{L}(\mathbf{x}_i, \hat{x}_i) = -\sum_{i=1}^{N} (1 - y_i)(\mathbf{x}_i - \hat{x}_i)^2 \tag{2}$$

The term $1 - y_i$ acts as a filter, considering only the normal samples ($y_i = 0$) in the loss calculation. $\mathcal{L}(\theta_1)$: This is the loss calculated using the predictions from the noise detection (ND) block. It evaluates the model's capacity to correctly classify data points as either normal or anomalous. It measures how well the ND block distinguishes between regular data points and outliers or anomalies in the dataset.

$$\mathcal{L}(\theta_1) = \sum_{i=1}^{N} \mathcal{L}(y_i, \hat{y}_i) = -\sum_{i=1}^{N} y_i \log(\hat{y}_i) + (1 - y_i) \log(1 - \hat{y}_i) \tag{3}$$

The overall loss function is constructed as a weighted sum of these two components, with the weight $w$ determining the relative importance assigned to each. The loss function is defined as follows:

$$\mathcal{L}(\theta) = w \cdot \mathcal{L}(\theta_1) + (1 - w) \cdot \mathcal{L}(\theta_0) \tag{4}$$

- $\theta = (\theta_0, \theta_1)$. $\theta_0$ and $\theta_1$ are respectively the parameters of the ND block and the AE block and $\theta$ represents the overall model parameters. N is the total number of samples. $\hat{y}_i$ is the estimated probability that sample $i$ is an anomaly calculated by the ND block.

- $0 \leq w \leq 1$ determines the relative importance assigned to each component of the loss function. The weight parameter $w$ plays a pivotal role in shaping the model's behavior. It acts as a tuning knob that adjusts the balance between the AE block's reconstruction capabilities and the ND block's noise detection capabilities within the loss function. When $w$ is set to a higher value, the ND blocks contribution is accentuated, indicating a greater focus on anomaly detection. This can be especially useful in situations where accurate anomaly identification is of paramount importance. In contrast, when $w$ is set lower, it places more emphasis on the autoencoder's ability to faithfully reconstruct the input data, which can be advantageous when the quality of the normal data reconstruction is a primary concern. When $w = 0$, $\mathcal{L}(\theta_1)$ is entirely ignored, and when $w = 1$, $\mathcal{L}(\theta_0)$ is ignored. The final loss can be rewritten as follows:

$$\mathcal{L}(\theta) = w \cdot \mathcal{L}(\theta_1) + (1 - w) \cdot \mathcal{L}(\theta_0) = w \sum_{i=1}^{N} \mathcal{L}(y_i, \hat{y}_i) + (1 - w) \sum_{i=1}^{N} \mathcal{L}(\mathbf{x}_i, \hat{x}_i)$$

$$= -\sum_{i=1}^{N} \left[ (1 - w) \cdot (1 - y_i)(\mathbf{x}_i - \hat{x}_i)^2 + w \left[ \cdot y_i \log(\hat{y}_i) + (1 - y_i) \log(1 - \hat{y}_i) \right] \right]$$

$$= -\sum_{i=1}^{N} \left[ w \cdot y_i \log(\hat{y}_i) + (1 - y_i) \left[ (1 - w) \cdot (\mathbf{x}_i - \hat{x}_i)^2 + w \cdot \log(1 - \hat{y}_i) \right] \right]$$

- $w \cdot y_i \log(\hat{y}_i)$: This term is associated with the noise detection (ND) block and focuses on the classification of data points as either anomalous ($y_i = 1$) or non-anomalous ($y_i = 0$). It calculates the logarithm of the predicted probability $\hat{y}_i$ for the true class labels $y_i$. When $y_i = 1$ (indicating

an anomaly), this term encourages $\hat{y}_i$ to be close to 1 indicating high confidence in the anomaly prediction. It measures how well the model's predictions $\hat{y}_i$ align with the true labels $y_i$. When $y_i = 1$, this term evaluates how well the model predicts the probability of an anomaly ($\hat{y}_i$).

- $(1 - y_i) \left[ (1 - w) \cdot (\mathbf{x}_i - \hat{x}_i)^2 + w \cdot \log(1 - \hat{y}_i) \right]$: This part combines contributions from both the autoencoder (AE) block and the ND block. It quantifies how each block contributes into the loss of a negative sample. This part combines contributions from both the autoencoder (AE) block and the ND block:

  $(1 - w) \cdot (\mathbf{x}_i - \hat{x}_i)^2$: This component reflects the reconstruction error from the AE block. When $y_i = 0$ (non-anomaly), it encourages the difference between the input data $\mathbf{x}_i$ and its reconstruction $\hat{x}_i$ to be minimized. This term drives the AE to capture meaningful data representations. When $w$ is closer to 1, this term contributes less to the loss, emphasizing data reconstruction.

  $w \cdot \log(1 - \hat{y}_i)$: This component is related to the anomaly detection objective. It encourages the logarithm of $(1 - \hat{y}_i)$ when $y_i = 0$. In other words, it encourages the model to assign lower probabilities to non-anomalous data points. It assesses how well the model predicts the probability of a non-anomaly $(1 - \hat{y}_i)$ when $y_i = 0$ (indicating a negative class or non-anomaly).

- **Reconstruction Quality:** Encouraged by the $(1-w) \cdot (\mathbf{x}_i - \hat{x}_i)^2$ term, this part of the loss function motivates the AE block to learn meaningful data representations. It measures how accurately the model can reconstruct input data when the data is non-anomalous ($y_i = 0$. A lower reconstruction error implies that the AE is successful in capturing essential features of the data.

- **Anomaly Detection**: Guided by the $w \cdot y_i \log(\hat{y}_i)$ and $w \cdot \log(1 - \hat{y}_i)$ terms, the ND block focuses on accurately classifying data points as anomalies or non-anomalies. It encourages the model to assign high probabilities ($\hat{y}_i$ close to 1) to anomalies ($y_i = 1$) and low probabilities ($\hat{y}_i$ close to 0) to non-anomalies ($y_i = 0$).

The objective during model training is to find the optimal configuration of model parameters $\hat{\theta}$ that minimizes the training loss $\mathcal{L}(\theta)$. The optimization objective is defined as follows:

$$\hat{\theta} = \arg \min_{\theta} \mathcal{L}(\theta) = \arg \min_{\theta} \sum_{i=1}^{N} \mathcal{L}(x_i, y_i, \theta) \tag{5}$$

## 4.4 ANOMALY SCORE

In the AnoRand model, we use two outputs: $\hat{y}$ from the ND block and $\hat{x}$ from the AE block's reconstruction, for classifying input samples as anomalies or normal data. We combine these outputs to calculate the final anomaly score. This leverages the model's ability to detect anomalies via the ND block's focus on data noise and to produce high-quality reconstructions using the autoencoder. We define $\hat{y}_0$ as the predicted probability of a sample being an anomaly based on the AE block reconstruction $\hat{x}$. Higher reconstruction errors indicate a higher likelihood of being an anomaly. To balance $\hat{y}$ and $\hat{y}_0$, we introduce a weight parameter $\alpha$, which adapts the model's behavior based on the dataset and the noise detection versus reconstruction trade-off. $\alpha$ is calculated from the third quantile of the anomaly score from both blocks. Specifically, $\alpha$ and $\hat{y}_0$ are determined as follows:

$$\hat{y}_0 = \frac{1}{1 + e^{-(x - \hat{x})^2}} \qquad\qquad \alpha = \frac{Q_3^1}{Q_3^0 + Q_3^1}$$

- $Q_3^0$ represents the third quantile of the autoencoder block's predicted probabilities $\hat{y}_0$.
- $Q_3^1$ represents the third quantile of the noise detection block's predicted probabilities $\hat{y}$.

Using quantiles, such as the third quantile, provides a robust measure for estimating the range of predicted probabilities. This approach is less sensitive to extreme values and outliers, making it suitable for ensuring that the weight $\alpha$ is derived from a reliable range of values. With the weight $\alpha$ determined, the final anomaly score denoted as $\hat{y}_{score}$ is computed as a weighted sum of $\hat{y}$ and $\hat{y}_0$:

$$\hat{y}_{score} = \hat{y} \cdot (1 - \alpha) + \alpha \cdot \hat{y}_0 \tag{6}$$

This final prediction formula allows the model to adaptively combine the strengths of the ND block and the AE block. When $\alpha$ is close to 0, the ND block's output has more influence, prioritizing noise detection. When $\alpha$ is close to 1, the AE block's output plays a more significant role, emphasizing autoencoder reconstruction. This dynamic adjustment ensures that the model can effectively handle various anomaly detection challenges.

## 5 EXPERIMENT

For all upcoming experiments, each dataset has been separated into a training dataset (80 %) and test (20 %) dataset. To avoid any risk of data leakage, models are trained using a k-fold cross-validation ($k = 10$ in our case). Each model is therefore trained and validated on 10 different subsamples. The final value of each performance metric is computed by taking the average of the ten measurements obtained during the ten iterations of the cross-validation. We choose the best hyperparameters using the library Omar et al. (2020) and on a small holdout set ($10\%$) of the validation dataset. We compared the performances of our method to those of 17 baseline unsupervised clustering algorithms available in the python Outlier Detection (PyOD) package Zhao et al. (2019).In the simulated experiments, we generated classification datasets using the make_classification module from sklearn. Each generated a training set has an imbalance rate of 2%. Note that for iteration in each experiment, we generated new samples by varying the random state parameter. We evaluate the algorithms using two widely used metrics: AUC ROC (Area Under Receiver Operating Characteristic Curve) and AUC PR (Area Under Precision-Recall Curve). In all incoming experiments, we report these two metrics as performance metrics. The higher the values, the better is the algorithm. In their article, Saito et al. Saito & Rehmsmeier (2015) showed that AUC ROC could be misleading when applied in imbalanced classification scenarios, instead AUC PR should be used. The experimental setting is described in details in the Appendix 8.2.

### 5.1 COMPARATIVE STUDY OF ANOMALY DETECTION ALGORITHMS

To ensure a robust evaluation, we divided the dataset into a training set and a test set, with 70% of the samples allocated to training and the remaining 30% reserved for testing. Recall that at this stage, only a small subse (2%) of the training data was used to create synthetic labels, as discussed in detail in subsection 4.2. This meticulous label generation process ensured that our model's performance remained independent of the specific samples chosen during label creation. We conducted a comprehensive comparative analysis, pitting our proposed method against 17 baseline unsupervised clustering algorithms. For a rigorous assessment of algorithmic performance, we performed multiple iterations of training and testing for each algorithm (10 times). During these iterations, synthetic labels were consistently derived from 2% of the training data to ensure fairness and impartiality.

Figure 2 and Figure 7 show that our proposed method consistently outperformed all other algorithms in terms of AUC PR and AUC ROC. When compared to deep learning-based unsupervised methods like Deep Autoencoder (AE), Variational Autoencoder (VAE), and Multi-Layer Perceptron (MLP), our approach demonstrated superior performance, despite sharing similar architectural foundations. Intriguingly, deep learning-based unsupervised methods, such as DeepSVDD and Autoencoder, exhibited unexpectedly subpar performance compared to classical techniques.

It is worth noting that while our method achieved good performance, it came at the cost of increased training time. In Figure 2b, we provide a visual representation of the comparison of training duration between the algorithms. The reasons behind this prolonged training time may be attributed to the complexity of our model, the number of training iterations and the number of parameters.

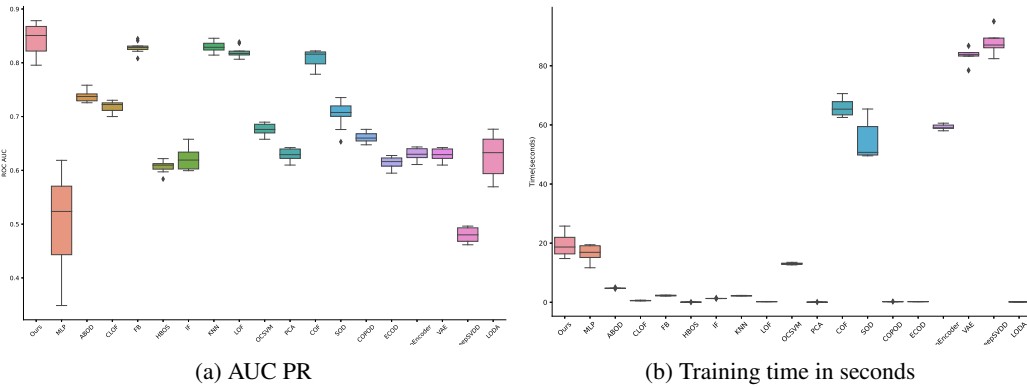

|     (a) AUC PR     |     (b) Training time in seconds     |

Figure 2: Performance metrics on synthetic data set for unsupervised algorithms

## 5.2 Enhanced Latent Representations: AnoRand vs. Traditional Autoencoder

In Figure 3, we delve into the latent representations extracted from the MNIST handwriting dataset, where we compare the outcomes achieved by a traditional deep autoencoder with those generated by our innovative AnoRand model. In our experiment using the MNIST dataset, we compared three models: a traditional deep autoencoder (AE), a variational autoencoder (VAE), and our novel AnoRand model. Notably, all three models shared a similar architecture, featuring a comparable number of layers and neurons. We focused on distinguishing the challenging digits 1 and 7, designating 2% of digit 7 samples as anomalies. The results from Figure 3d indicate that AE and VAE outperform AnoRand in image reconstruction due to their broader learning objectives. However, the results in Figure 8(in Appendix) demonstrate that AnoRand excels in anomaly detection, with fewer false negatives, thanks to its specialized focus on the normal class. Visual inspection of the latent vectors, as depicted in Figure 3 confirms AnoRand's ability to distinguish between classes, while AE and VAE struggle in this regard. Overall, AnoRand demonstrates superior performance in crafting latent representations for anomaly detection, surpassing traditional autoencoders.

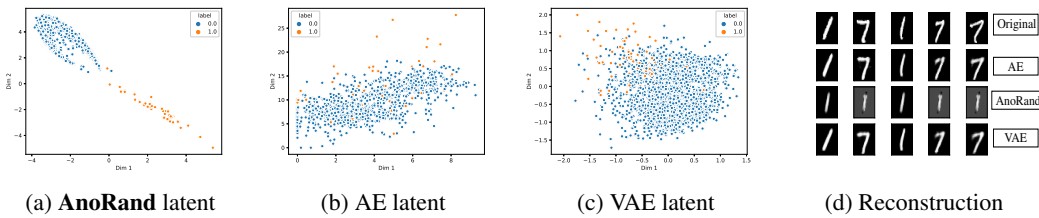

(a) **AnoRand** latent     (b) AE latent     (c) VAE latent     (d) Reconstruction

Figure 3: **AnoRand** compared to traditional autoencoders

## 6 Unsupervised anomaly detection on real world datasets

We compared the performance of our method (**AnoRand**) to those of state-of-the-art unsupervised methods on the ADBench anomaly detection benchmark Han et al. (2022). In their paper, they compared the performance of 17 algorithms on 57 benchmark datasets (see Table 3 in Appendix for full data description). In our experiments, we choose 44 of the most difficult datasets to compare the algorithms. The datasets cover different fields including healthcare, security and more. We grouped the datasets into four categories to make the comparison easy: Image datasets, NLP datasets, Healthcare datasets, Science datasets and datasets from other fields (documents, web etc.). In their paper, the authors compared supervised, semi-supervised and unsupervised anomaly detection methods on these datasets. In our study, we only focus on the semi-supervised and unsupervised algorithms of the benchmark. We reported the average performances over teen different iterations. In Table 1, Table 2 and Figure 9, we report the algorithms performance and their rankings on the ADBench real-world datasets. In Figure 4a and 9a, the boxplots show that our model has the best overall ranking among all its counterpart unsupervised algorithms. Indeed, Figure 4b shows that in terms of AUC PR, AnoRand is ranked first ($1^{st}$) on 20 datasets, second ($2^{nd}$) on 5, third ($3^{rd}$) on 6 and fourth ($4^{th}$) on 3. Figure 9b on the other hand shows that in terms of AUC ROC, AnoRand is ranked first ($1^{st}$) on 18 datasets, second ($2^{nd}$) on 3, third ($3^{rd}$) on 6 and fourth ($4^{th}$) on 4. The results also show that, in situations where another algorithm outperforms ours, the performance gap is very small in most cases. The results and performances (AUC ROC and AUC PR) of all experiments are fully reported in Table 1 (AUC PR values) and Table 2 (AUC ROC values) in the Appendix.

**Results on image classification data sets.** This category encompasses diverse datasets curated for computer vision tasks, including image classification (e.g., Mnist, CIFAR10, and MVTec). Pre-trained ResNet18 models He et al. (2016) have been used to extract data embedding from the original images. AnoRand, our algorithm, outperforms all reference algorithms on 6 of 11 image datasets, demonstrating its robustness and adaptability (see Table 1 and 2 in Appendix). It achieves the second-best performance on 2 additional datasets, highlighting its versatility.

**Results on NLP data sets.** The natural language processing (NLP) data sets consist of text data and are vital for training and evaluating models that handle language-related tasks, including text classification, sentiment analysis, named entity recognition, machine translation and more. These datasets often comprise a collection of text documents, sentences or tokens, each associated with specific la-

bels or annotations. For these datasets, they used a BERT (Bidirectional Encoder Representations from Transformers) Devlin et al. (2019) pre-trained on the BookCorpus and English Wikipedia to extract the embedding of the token. In Table 1 and 2, we report the performances of anomaly detection algorithms in the NLP datasets. On these datasets, AnoRand outperforms the other algorithms on the speech and Imdb dataset, showcasing its remarkable ability to distinguish outliers in spoken and written language. Even in scenarios where it does not clinch the top spot, it consistently holds its own, securing a respectable third place on the remaining datasets. The algorithm's performance in the NLP benchmark datasets underscores its versatility and effectiveness in grappling with the nuances of linguistic data.

**Results on Healthcare data sets.** Table 1 and 2 (in Appendix) shows the algorithms performance on the 10 healthcare benchmark datasets. These datasets encapsulate a diverse array of medical data, ranging from Hepatitis detection to cancer diagnosis, each carrying profound implications for patient care and health monitoring. These results show that AnoRand consistently outperforms its algorithmic counterparts on a significant 40% of the benchmark datasets. Moreover, AnoRand has the best overall ranking across all ten healthcare datasets, demonstrating its effectiveness in addressing the unique challenges posed by healthcare data. Whether it's identifying anomalies in patient records or clinical observations, AnoRand proves to be a promising and robust solution.

**Results on Other datasets.** In this category, we include all other datasets from other fields such as experiment science, Sociology, Botany, Finance etc. The inclusion of datasets from such disparate fields underscores the universality of anomaly detection as a critical analytical tool. On these datasets, AnoRand outperforms the other algorithms on 10 out of 19 datasets. Moreover, even in cases where AnoRand doesn't achieve the best performance, it maintains its competitive edge, securing a position no lower than fourth on six(6) additional datasets.These results underscore the algorithm's potential to address anomalies across a wide spectrum of scenarios, ranging from scientific experiments to social phenomena and financial markets.

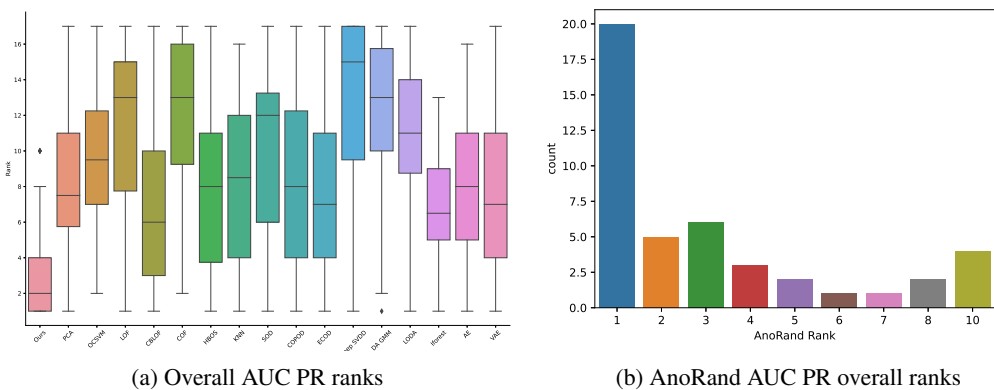

(a) Overall AUC PR ranks         (b) AnoRand AUC PR overall ranks

Figure 4: Algorithms performance rankings on real-world data sets (lower the better).

## 7 CONCLUSION

In this paper, we proposed a new semi supervised anomaly detection method based on deep autoencoder architecture. This new method that we called **AnoRand**, jointly optimizes the deep autoencoder and the FFP model in an end-to-end neural network fashion. Our method is performed in two steps: we first create synthetic anomalies by randomly adding noise to few samples from the training data; secondly, we train our deep learning model in supervised way with the new labeled data. Our method takes advantage of these limitations of FFP models in case of imbalance classes and use them to reinforce the autoencoder capabilities. Our experimental results show that our method achieves state-of-the-art performance on synthetic datasets and 57 real-world datasets, significantly outperforming existing unsupervised alternatives. Moreover, on most benchmark datasets, regardless of the category, AnoRand outperforms all its deep learning-based counterparts. The main limitation of our method is that the training time is longer compared to most state-of-the-art non-deep learning algorithms, although it remains shorter than that of deep learning algorithms.

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

# 8 APPENDIX

## 8.1 SYNTHETIC LABEL GENERATION

Recall that to build our semi-supervised model, we created the synthetic anomalies. In the initial stage, we randomly select a subset of samples from the training dataset that we denoted by $X_0$. These specially selected samples are designated as $X_0^0$. The remaining samples, those not included in $X_0^0$, are referred to as $X_1$. To transform $X_0^0$ into synthetic anomalies, we introduce noise to these samples, resulting in a new set named $X_0^1$. The addition of noise helps us simulate the variability and unpredictability often found in real-world anomaly data. The choice of noise generation technique plays a crucial role in the quality of synthetic anomalies. In our experiments, we opted for a hybrid approach, combining Gaussian noise with the Synthetic Minority Over-sampling Technique (SMOTE), a well-established method in the field of anomaly detection. This technique was introduced by Chawla et al.Chawla et al. (2002) and has proven to be one of the best techniques for synthesizing samples of the minority class in the presence of unbalanced data. The core principle of SMOTE is to generate synthetic samples by selecting a sample from the minority class and identifying its k nearest neighbors. For each selected neighbor, a synthetic sample is created by computing the difference between the feature vectors of the chosen sample and its nearest neighbor. This difference is then scaled by a random value between 0 and 1 and added to the feature vector of the chosen sample. In our experiments, we applied the SMOTE technique to $X_0^0$, the subset originally designated as anomalous. SMOTE was applied to oversample this subset until it constituted 5% of the total training dataset. To ensure that our synthetic anomalies closely resemble the feature distribution of normal samples, we thoughtfully merged the SMOTE technique with Gaussian noise. This hybrid approach ensures that the synthetic anomalies are not too dissimilar from the normal samples in terms of their feature distribution. This reflects the nuanced nature of anomalies in practical applications and enhances the model's ability to identify them accurately. Indeed, in real-world scenarios, anomalies are often subtle variations or "broken" versions of normal data. Our ultimate training dataset, denoted as $X_{tr}$, comprises three fundamental components: $X_1$, representing the majority of normal samples; $X_0^0$, initially serving as the foundation for synthetic anomalies; and $X_0^1$, the noisy, synthetic anomalies created through the combination of SMOTE and Gaussian noise. It is worth highlighting that this composition ensures that $X_{tr}$ now contains 5% of "anomalous" samples, encompassing both the originally selected $X_0^0$ samples and the synthetic anomalies generated through our innovative approach.

## 8.2 DETAILS ON EXPERIMENT SETTING

For all upcoming experiments, each dataset has been separated into a training dataset (80 %) and test (20 %) dataset. To avoid any risk of data leakage, models are trained using a k-fold cross-validation ($k = 10$ in our case). Each model is therefore trained and validated on 10 different sub-samples. The final performance is computed on the test set. Note that the test sets do not intervene at any time in the training steps. The final value of each performance metric is computed by taking the average of the ten measurements obtained during the ten iterations of the cross-validation. The final models are compared using the AUC (PRC) metric. We set the number of epochs to 50 for each model and used early stopping to avoid overfitting. We used the version of the mini-batch stochastic gradient descent (SGD) called SGD Adam with a batch size of 256. We kept the default values of the other hyperparameters: lr=0.001, $\beta_1 = 0.9$ and $\beta_1 = 0.999$. The rectified linear unit (ReLU) activation function is used in neurons of the hidden layers, and the sigmoid activation function is used for output layer to estimate posterior probabilities. We choose the best hyperparameters using the **KerasTuner** library and on a small holdout set ($10\%$) of the validation dataset. KerasTuner is an easy-to-use, scalable hyperparameter optimization framework that solves the pain points of hyperparameter search Omar et al. (2020). We also added between the hidden layers a Batch Normalization layer followed by a dropout layer.

For the state-of-the-art algorithms, we used their implementation in the python Outlier Detection (PyOD) package Zhao et al. (2019). PyOD package is an open-source, comprehensive, and widely used toolkit for detecting outliers in multivariate data. It provides a collection of algorithms and tools to identify and analyze these outliers across various application domains. PyOD contains a wide range of outlier detection algorithms, both traditional and state-of-the-art. These algorithms

cover various statistical, distance-based and machine learning-based approaches. PyOD is built with efficiency in mind, enabling the processing of large datasets with complex structures.

In these experiments, we simulated a classification dataset using the make_classification module from sklearn. The make_classification module creates clusters of samples normally distributed about the vertices of a hypercube and assigns an equal number of clusters to each class. It then introduces interdependence between the created features and adds various types of noise to the data. We generated a training set of 20000 samples with an imbalance rate of 2%. This means that the minority class represents 2% of the training dataset. Note that for iteration in each experiment, we generated new samples by varying the random state parameter.

## 8.3 CHOICE OF THE OPTIMAL VALUE FOR $w$.

Recall that $w$ is the weight assigned to the loss of the noise detection block. We hypothesize that when $w$ tends to 1, the influence of the autoencoder block tends to 0 and the final model is equivalent to a simple feed-forward perceptron (FFP) model. So, by varying the weights, we expect to see the impact of each part of our architecture to the final model loss. For each value of $w \in [0, 1]$, we trained our model 10 times on 10 different samples and report its AUC PR in figure 5a and the AUCROC in figure 5b. The figures show that the model performance increases until 0.2 and decreases very fast when $w$ is greater than 0.2. The boxplot at 0.2 shows that the model's AUC PR and AUCROC are stable. Indeed, at this point, the interquartile ranges of the boxplots are small and there are fewer outliers. These results suggest that, in the proposed architecture, the ND block positively contributes to the performance of the final model up to a certain level. The optimal value of $w$ lays around 0.2.

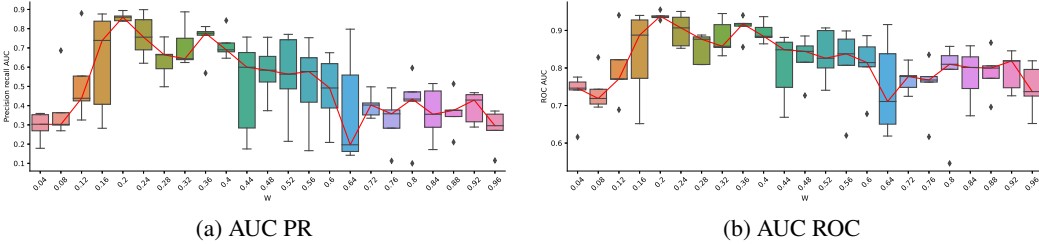

(a) AUC PR                                    (b) AUC ROC

Figure 5: Performance metrics by varying w

## 8.4 REMOVING SOME NEURONS FROM THE ND BLOCK LATENT

In this subsection, we aim to assess the impact of the noise detection (ND) block's high-level embedding $z_1 = F(x, \theta_0)$ within our trained model. Our primary objective is to gain a deeper understanding of the significance of the ND block and its role in shaping the model's overall performance. To achieve this goal, we begin with a fundamental hypothesis: if the ND block doesn't play a crucial role or doesn't significantly contribute to the autoencoder block's performance, then it should be possible to set its latent embedding vector to zeros without substantially diminishing the model's overall performance. In other words, we test whether our proposed architecture adds value beyond a simple autoencoder. Our experimental setup involves training a model according to the architecture we have designed. We then proceed to selectively zero out certain neurons within the last layer of the ND block. The number of neurons to be zeroed out is represented by the variable $\beta$.

In our ND block, there are a total of $N_{latent}$ neurons within the last layer. To determine the number of unique combinations for selecting $\beta$ neurons out of $N_{latent}$, we employ the combination formula $C(N_{latent}, \beta)$, yielding $N_0$ possible combinations.

$$N_0 = C(N_{latent}, \beta) = \frac{N_{latent}!}{\beta!(N_{latent} - \beta)!}$$

For each specific value of $\beta$, we randomly generate a set of 100 different combinations of neurons to set to zeros within the ND block. The performance of our model is then evaluated under each

configuration. Figures 6a and 6b show the variation in performance of the model. The plotted figures reveal a notable trend: as the number of neurons within the ND block set to zero ($\beta$)increases, the model's performance decreases exponentially. This observation strongly supports the hypothesis that our proposed architecture, with the ND block intact, significantly enhances the model's learning.

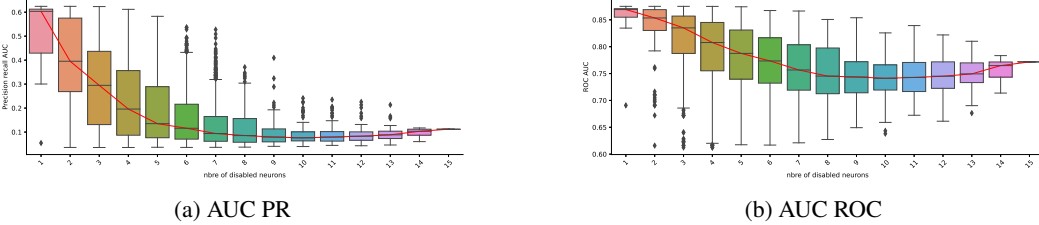

(a) AUC PR        (b) AUC ROC

Figure 6: Performance metrics by disabling some neurons

## 8.5 ENHANCED LATENT REPRESENTATIONS: ANORAND VS. TRADITIONAL AUTOENCODER

Within this experimental framework, we constructed three distinct models: an AE, a variational AE, and our novel AnoRand. Notably, all three models shared a similar architecture, featuring a comparable number of layers and neurons, and were implemented within an unsupervised learning framework. Our results show that when it comes to image reconstruction, both AE and VAE showed superior capabilities (see figure 3d). This phenomenon can be attributed to their inherent nature, which drives them to learn representations of all classes present in the data. In contrast, AnoRand, by design, primarily focuses on a single class during training, akin to a one-class model. Consequently, it excels at reconstructing instances from the normal class (digit 1) but faces limitations when dealing with digit 7, which it considers as an anomaly. However, the results in figure 8 demonstrate that AnoRand surpasses AE and VAE in anomaly detection tasks. The primary reason behind this performance differential lies in AnoRand's exclusive focus on learning the normal class. This specialized training equips AnoRand to effectively spot data points from the anomaly class, thereby minimizing false negatives. In contrast, AE and VAE, due to their broader learning objectives, exhibit a higher incidence of false positives, primarily because data from the anomaly class produces reconstructions very similar to those of the normal class.

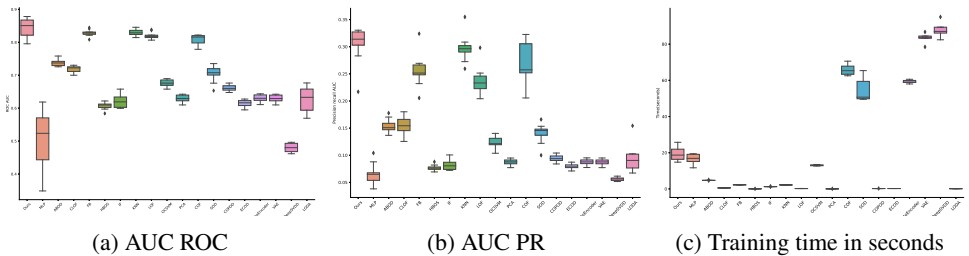

(a) AUC ROC      (b) AUC PR      (c) Training time in seconds

Figure 7: Performance metrics on synthetic data set for unsupervised algorithms

## 8.6 UNSUPERVISED ANOMALY DETECTION ON REAL WORLD DATASETS

We compared the performances of our method to those of 16 baseline unsupervised clustering algorithms including: CBLOF He et al. (2003), HBOS Goldstein & Dengel (2012), KNNRamaswamy et al. (2000), IForest Liu et al. (2008), LOF Breunig et al. (2000), OCSVM Schölkopf et al. (1999), PCAShyu et al. (2003), COFTang et al. (2002), SOD Kriegel et al. (2009), COPOD Li et al. (2020), ECOD Li et al. (2022), AutoEncoder Kingma & Welling (2013), DeepSVDD Ruff et al. (2018), GMM Zong et al. (2018) and LODAPevnỳ (2016).

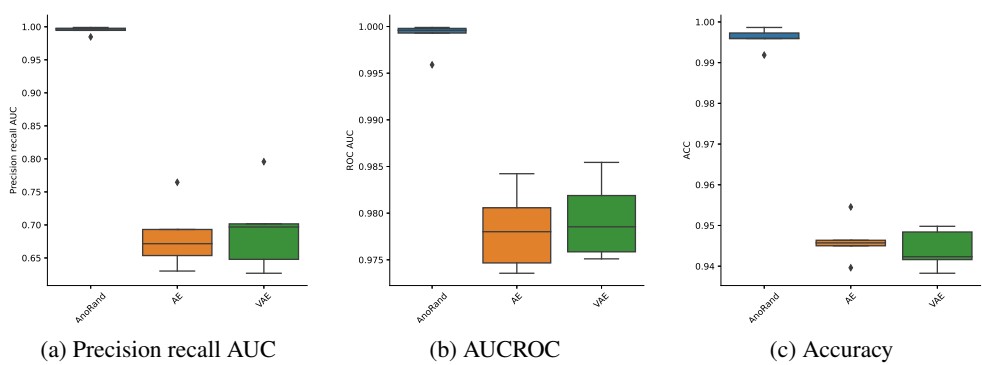

(a) Precision recall AUC      (b) AUCROC      (c) Accuracy

Figure 8: Performance of **AnoRand** compared to autoencoders on Mnist 2 classes data

Table 1: AUCPR (in %) of 16 unsupervised algorithms on 44 real-world datasets. For our method, we computed the average AUCPR over 10 runs and added the standard deviation. The performance rank is shown in parentheses (the lower, the better), and mark the best performing method(s) in **bold**.

| Category | Dataset | PCA | OCSVM | LOF | CBLOF | COF | HBOS | KNN | SOD | COPOD | ECOD | Deep SVDD | DA GMM | LODA | Iforest | AutoEncoder | VAE | AnoRand(Ours) |
|---|---|---|---|---|---|---|---|---|---|---|---|---|---|---|---|---|---|---|
| Image and CV | mnist | 39.93(3) | 33.2(6) | 20.9(14) | 28.82(8) | 25.51(11) | 12.51(17) | 35.53(5) | 19.15(16) | 21.35(13) | 31.93(7) | 19.72(15) | 23.75(12) | 25.86(10) | 27.71(9) | 40.28(2) | 39.87(4) | **50.17 ±1.25 (1)** |
| | optdigits | 2.76(14) | 2.92(13) | 6.06(4) | 10.08(2) | 4.42(7) | 10.03(3) | 3.06(12) | 4.39(8) | 4.36(9) | 3.43(11) | - | 5.59(5) | 3.95(10) | 5.09(6) | 2.64(16) | 2.67(15) | **56.34 ±5.42(1)** |
| | skin | 17.4(14) | 19.03(9) | 18.25(12) | 29.82(2) | 16.38(15) | 23.7(8) | 28.72(4) | 24.61(7) | 17.99(13) | 15.96(16) | 18.48(10) | - | 18.44(11) | 26.08(6) | 28.08(5) | 29.78(3) | **52.20 ±12.26(1)** |
| | FashionMNIST | 31.42(9) | 31.97(8) | 16.85(16) | 38.9(2) | 20.73(14) | 29.43(11) | 33.87(3) | 28.72(12) | 30.32(10) | 32.53(5) | 17.43(15) | 14.44(17) | 27.32(13) | 32.35(7) | 32.46(6) | 32.66(4) | **43.16 ±2.48(1)** |
| | MNIST-C | 16.88(10) | 17.72(7) | 13.84(15) | 27.62(2) | 14.53(14) | 15.46(13) | 22.98(3) | 15.68(12) | 15.9(11) | 18.24(5) | 8.34(17) | 11.37(16) | 18.63(4) | 17.99(6) | 17.03(9) | 17.24(8) | **35.21 ±1.26(1)** |
| | satimage-2 | 85.69(6) | 82.71(7) | 4.29(16) | 97.09(1) | 8.81(15) | 78.04(9) | 39.14(12) | 26.11(13) | 76.55(10) | 63.25(11) | 3.08(17) | 22.07(14) | 80.52(8) | 93.45(3) | 86.36(5) | 87.52(4) | 94.12 ±0.80(2) |
| | MVTec-AD | 54.06(9) | 51.44(13) | 54.9(7) | 58.52(1) | 46.59(15) | 55.22(6) | 55.55(4) | 51.48(12) | 54.64(8) | 55.44(5) | 36.5(17) | 45.66(16) | 49.73(14) | 56.04(3) | 51.57(11) | 51.62(10) | 57.65 ±0.12(2) |
| | letter | 6.86(15) | 6.11(17) | 34.02(1) | 14.8(6) | 21.43(5) | 8.38(10) | 30(2) | 28.63(3) | 6.77(16) | 6.94(13) | 9.29(8) | 11.68(7) | 6.87(14) | 8.49(9) | 8.25(12) | 8.31(11) | 28.47 ±1.84(4) |
| | celeba | 15.89(1) | 10.73(8) | 1.71(17) | 11.33(7) | 1.77(16) | 13.82(2) | 3.14(12) | 2.66(13) | 13.69(3) | 12.37(4) | 2.34(14) | 1.95(15) | 4.04(11) | 8.96(9) | 11.39(5) | 11.39(5) | 5.61 ±0.50(10) |
| | CIFAR10 | 10.59(6) | 10.19(9) | 13.02(1) | 10.61(5) | 11.61(2) | 8.38(15) | 11.13(3) | 11.06(4) | 8.77(14) | 9.29(12) | 8.05(16) | 7.73(17) | 9.72(11) | 8.97(13) | 10.32(8) | 10.45(7) | 10.04 ±0.91 (10) |
| NLP | speech | 1.97(13) | 1.96(14) | 2.52(3) | 1.99(12) | 2.25(5) | 2.09(9) | 2.02(11) | 2.13(8) | 1.94(15) | 1.77(17) | 5.12(2) | 2.03(10) | 1.79(16) | 2.31(4) | 2.16(6) | 2.16(6) | **7.65 ±0.11 (1)** |
| | Imdb | 4.55(15) | 4.44(17) | 4.83(8) | 4.75(9) | 5.16(4) | 4.74(10) | 4.49(16) | 4.7(12) | 4.9(6) | 4.9(6) | 5.06(5) | 4.65(13) | 4.59(14) | 4.74(10) | 6.18(3) | 6.3(2) | **7.65 ±0.25 (1)** |
| | Agnews | 5.74(11) | 5.69(12) | 14.35(1) | 7.02(6) | 12.21(2) | 5.58(13) | 8.61(4) | 8.4(5) | 5.43(14) | 5.43(14) | 4.45(17) | 5.41(16) | 5.93(10) | 6.04(9) | 6.18(8) | 6.3(7) | 9.04 ±1.58 (3) |
| | Amazon | 5.85(10) | 5.64(16) | 5.72(13) | 6.07(5) | 5.74(12) | 5.98(7) | 6.23(2) | 6.4(1) | 6.08(4) | 6.06(6) | 6.39(17) | 6.72(16) | 5.92(9) | 5.95(8) | 5.75(11) | 5.69(14) | 6.20 ±0.23 (3) |
| | Yelp | 7.62(13) | 7.75(10) | 8.52(5) | 7.68(11) | 8.68(4) | 7.81(9) | 9.85(1) | 9.2(2) | 8.01(6) | 7.98(7) | 6.39(17) | 6.72(16) | 7.65(12) | 7.88(8) | 7.05(15) | 7.14(14) | 8.80 ±1.32 (3) |
| Healthcare | WBC | 82.29(9) | 89.87(6) | 5.57(16) | 92.27(4) | 9.73(14) | 73.56(11) | 66.55(12) | 54(13) | 86.19(7) | 86.19(7) | 6.38(15) | #N/A | 78.67(10) | 90.49(5) | 94.3(3) | 94.45(2) | **94.55 ±4.66(1)** |
| | Cardiotocography | 47.95(6) | 52.61(2) | 30.66(14) | 45.44(7) | 28.21(16) | 38.28(11) | 34.79(12) | 27.99(17) | 40.46(10) | 43.57(8) | 34.03(13) | 30.61(15) | 48(5) | 41.47(9) | 48.49(3) | 48.16(4) | **61.39 ±5.74(1)** |
| | Lymphography | 97.02(6) | 93.59(7) | 23.08(14) | 97.62(2) | 36.68(13) | 91.83(8) | 38.69(12) | 22.65(15) | 88.68(10) | 90.87(9) | 4.58(17) | 19.52(16) | 44.54(11) | 97.31(3) | 97.22(5) | 97.24(4) | **99.68 ±0.01(1)** |
| | breastw | 95.11(7) | 82.7(13) | 28.55(15) | 91.54(10) | 27.6(16) | 97.71(4) | 92.19(9) | 84.88(12) | 99.4(1) | 98.54(2) | 50.92(14) | - | 97.04(5) | 96.04(6) | 89.17(11) | 94.96(8) | 98.17 ±0.40(3) |
| | WPBC | 23.01(8) | 22.93(9) | 20.29(17) | 21.32(15) | 21.3(16) | 23.04(7) | 21.49(13) | 25.37(3) | 22.81(10) | 21.38(14) | 26.24(1) | 22.49(11) | 25.58(2) | 22.42(12) | 23.91(4) | 23.44(5) | 23.06 ±0.44(6) |
| | Hepatitis | 36.65(4) | 29.44(10) | 13.69(17) | 31.54(8) | 14.39(16) | 37.73(3) | 21.95(15) | 24.89(12) | 41.5(1) | 37.82(2) | 22.17(14) | 22.96(13) | 30.9(9) | 26.25(11) | 33.81(6) | 33.58(7) | 35.20 ±7.17(5) |
| | thyroid | 44.34(6) | 21.23(12) | 20.81(13) | 29.95(9) | 28.5(10) | 50.98(3) | 34.98(8) | 23.56(11) | 19.64(14) | 54.05(2) | 2.5(17) | 16.06(15) | 14.68(16) | 63.11(1) | 46.16(5) | 36.9(7) | 47.79 ±2.77(4) |
| | aimthyroid | 16.12(11) | 10.37(15) | 15.71(12) | 13.69(14) | 14.39(13) | 16.99(8) | 16.74(9) | 18.84(6) | 16.58(10) | 24.65(2) | 21.95(4) | 9.64(16) | 7.06(17) | 30.47(1) | 22.73(3) | 19.4(5) | 17.02 ±0.80(5) |
| | Pima | 54.03(5) | 50(8) | 47.18(11) | 53.19(6) | 44.7(13) | 56.61(1) | 55.14(4) | 48.24(10) | 55.19(3) | 37.3(16) | 35.87(17) | 41.55(15) | 44.09(14) | 55.82(2) | 46.56(12) | 49.53(9) | 50.09 ±0.89(7) |
| | cardio | 66.06(2) | 62.89(5) | 23.79(16) | 61.95(6) | 28.67(14) | 52.1(11) | 40.72(12) | 28.54(15) | 60.42(7) | 68.59(1) | 22.5(17) | 28.92(13) | 53.41(9) | 59.95(8) | 64.72(3) | 62.9(4) | 52.53 ±4.68 (8) |
| Others | musk | 99.89(4) | 10.61(11) | 2.82(15) | 100(1) | 2.61(16) | 100(1) | 9.65(12) | 7.59(13) | 34.79(9) | 34.95(8) | 5.39(14) | 32.75(10) | 47(7) | 99.61(6) | 100(1) | 99.89(4) | **100 ±0(1)** |
| | Waveform | 5.79(12) | 4.37(16) | 11.33(5) | 18.98(2) | 14.11(3) | 5.86(11) | 13.04(4) | 9.6(6) | 6.9(7) | 6.8(8) | 4.83(13) | 3.11(17) | 4.71(14) | 6.2(9) | 6.2(9) | 4.65(15) | **33.26 ±0.79(1)** |
| | cover | 9.8(7) | 11.41(5) | 8.1(9) | 5.83(15) | 4(16) | 6.83(13) | 6.16(14) | 3.88(17) | 11.37(6) | 15.63(3) | 8.1(9) | 27.59(2) | 13.06(4) | 8.8(8) | 7.27(11) | 7.25(12) | **34.19 ±0.54(1)** |
| | fault | 32.76(14) | 38.44(8) | 38.38(9) | 43.98(4) | 41.56(5) | 36.47(10) | 54.45(2) | 48.01(3) | 30.54(17) | 30.82(16) | 39.15(7) | 33.48(13) | 31.03(15) | 41.09(6) | 34.58(11) | 34.46(12) | **63.29 ±1.51 (1)** |
| | donors | 17(6) | 9.8(11) | 7.88(14) | 6.89(15) | 8.8(13) | 23.36(2) | 14.75(7) | 9.69(12) | 21.58(4) | 14.17(8) | 6.38(16) | 10.53(10) | 3.78(17) | 12.74(9) | 22.74(3) | 18.78(5) | **90.85 ±6.83(1)** |
| | PageBlocks | 51.71(5) | 49.14(8) | 39.64(13) | 49.65(7) | 41.02(12) | 33.32(16) | 45.39(11) | 37.83(14) | 37.65(15) | 49(9) | 31.45(17) | 53.25(3) | 51.29(6) | 46.04(10) | 59.18(2) | 51.96(4) | **65.26 ±11.50(1)** |
| | magic.gamma | 59.27(7) | 51.43(15) | 54.76(12) | 68.85(3) | 54.12(14) | 62.41(6) | 75.63(2) | 67.89(4) | 59.18(8) | 54.38(13) | 49.17(16) | 46.92(17) | 58.49(11) | 64.72(5) | 59.18(8) | 59.11(10) | **77.93 ±1.29(1)** |
| | fraud | 22.91(11) | 47.58(2) | 47(4) | 47.52(3) | 22.86(12) | 25.89(10) | 47(4) | 31.37(9) | 42.82(8) | 42.99(7) | 8.97(17) | 21.32(14) | 46.37(6) | 21.67(13) | 15.88(15) | 15.88(15) | **60.08 ±0.15(1)** |
| | vertebral | 10.49(11) | 10.94(9) | 14.24(3) | 11.58(6) | 13.85(4) | 9.23(16) | 10.57(10) | 11.79(5) | 8.89(17) | 11.24(7) | 10.49(11) | 15.24(2) | 9.68(15) | 10.46(13) | 11.18(8) | 9.85(14) | **20.47 ±0.11(1)** |
| | SpamBase | 41.57(7) | 40.12(12) | 35.16(16) | 41.18(11) | 34.73(16) | 50.03(5) | 41.42(8) | 40.03(13) | 56.68(1) | 53.95(3) | 42.23(6) | - | 35.88(14) | 51.75(4) | 41.21(10) | 41.42(8) | 55.17 ±0.50(2) |
| | landsat | 16.18(17) | 16.21(16) | 24.69(7) | 30.97(3) | 24.95(6) | 22.03(10) | 24.65(8) | 26.38(4) | 17.48(14) | 25.17(5) | **38.83(1)** | 24.48(9) | 18.86(13) | 19.81(12) | 16.75(15) | 20.58(11) | 38.54 ±4.54 (2) |
| | shuttle | 92.35(7) | 85.29(10) | 13.76(16) | 60.98(11) | 12.17(17) | 96(4) | 20.38(13) | 20.27(14) | 96.56(2) | 95.76(5) | 15.86(15) | 93.2(6) | 48.75(12) | 91.27(9) | 91.49(8) | **97.62(1)** | 96.52 ±0.05(3) |
| | Stamps | 41.09(4) | 31.39(11) | 21.29(14) | 23.66(12) | 16.5(16) | 35.24(9) | 23.53(13) | 20.28(15) | 43(2) | 38.17(7) | 11.4(17) | **43.72(1)** | 34(10) | 39.49(5) | 38.35(6) | 38.11(8) | 34.19 ±0.39(3) |
| | satellite | 59.64(9) | 57.61(11) | 37.68(17) | 61.48(8) | 39.7(16) | 67.25(2) | 50.01(13) | 47.23(14) | 56.58(12) | 65.94(4) | 40.11(15) | 58.33(10) | 61.94(6) | 65.92(5) | 61.7(7) | 67.24(3) | **73.24 ±0.49(1)** |
| | wine | 30.87(5) | 21.56(8) | 7.77(16) | 5.83(17) | 8.45(13) | 43.08(3) | 8.43(14) | 7.95(15) | 45.71(2) | 18.37(11) | 21.14(9) | 17.51(12) | **48.82(1)** | 25.96(6) | 19.1(10) | 22.02(7) | 41.59 ±1.76(4) |
| | campaign | 27(9) | 29.22(5) | 14.51(14) | 23.99(10) | 13.01(16) | 37.99(2) | 27.18(8) | 18.88(12) | **38.58(1)** | 37(3) | 11.6(17) | 14.62(13) | | | 28.51(6) | 25.49(10) | 23.99 ±1.76 (10) |
| | http | 56.4(3) | 46.86(5) | 3.82(14) | 47.53(4) | 9.57(12) | 44.79(6) | 0.7(15) | 8.32(13) | 35.19(7) | 16.61(11) | 29(8) | | 0.67(16) | **90.83(1)** | 25.49(10) | 25.55(9) | 60.24 ±0.07(2) |
| | InternetAds | 32.55(11) | 54.68(2) | 40.49(9) | **58.13(1)** | 38.67(10) | 53.97(3) | 43.23(7) | 27.69(15) | 50.97(5) | 51.07(4) | 27.91(12) | 23.89(16) | 47.53(14) | 46.37(15) | 21.67(11) | 27.72(14) | 27.73(13) | 42.79 ±1.43 (8) |
| | census | **10.02(1)** | 6.76(12) | 5.45(13) | 7.4(9) | 4.88(16) | 8.6(6) | 9(4) | 8.5(7) | 9.9(2) | 9.7(3) | 6.87(11) | 8.7(5) | 5.01(15) | 7.7(8) | 5.41(14) | 4.45(17) | 7.31 ±0.10 (10) |

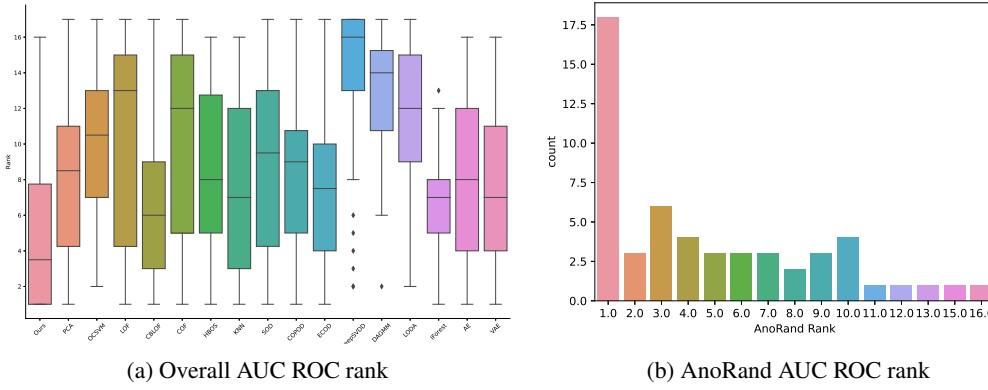

(a) Overall AUC ROC rank      (b) AnoRand AUC ROC rank

Figure 9: Algorithms performance rankings on real-world data sets (lower the better).

Table 2: AUC ROC of algorithms on various datasets. For our method, we computed the average AUC ROC over 10 runs and added the standard deviation. The performance rank is shown in parentheses (the lower, the better), and mark the best performing method(s) in **bold**.

| Datasets | PCA | OCSVM | LOF | CBLOF | COF | HBOS | KNN | SOD | COPOD | ECOD | Deep SVDD | DA GMM | LODA | IForest | AutoEncoder | VAE | AnoRand(Ours) |
|---|---|---|---|---|---|---|---|---|---|---|---|---|---|---|---|---|---|
| ALOI | 56,65(6) | 55,85(8) | **66,63(1)** | 55,22(9) | 64,68(2) | 52,63(14) | 61,47(3) | 61,09(4) | 53,75(12) | 56,6(7) | 50,29(17) | 51,96(15) | 51,33(16) | 56,66(5) | 54,96(10) | 54,89(11) | 53,42±2,89(13) |
| annthyroid | 66,25(11) | 57,23(15) | 70,2(7) | 62,26(13) | 65,92(12) | 60,15(14) | 71,69(6) | 77,38(3) | 76,8(4) | 78,03(2) | 76,62(5) | 56,53(16) | 41,02(17) | **82,01(1)** | 69,17(8) | 67,45(10) | 67,93±9,69(9) |
| backdoor | 80,13(9) | 86,2(4) | 85,68(5) | 81,16(6) | 73,03(10) | 71,43(12) | 80,82(8) | 60,54(13) | 80,97(7) | 86,33(3) | 55,16(17) | 56,26(16) | 69,22(14) | 72,15(11) | **88,7(1)** | 88,65(2) | 64,67±3,78(15) |
| breastw | 95,13(10) | 80,3(13) | 40,61(15) | 96,81(8) | 38,84(16) | 98,94(4) | 97,01(7) | 93,97(11) | **99,68(1)** | 99,17(2) | 65,66(14) | | 98,49(5) | 98,32(6) | 90,02(12) | 95,31(9) | 99,11±0,87(3) |
| campaign | 72,78(6) | 65,52(12) | 58,85(13) | 66,61(11) | 57,26(14) | **78,61(1)** | 72,1(7) | 69,04(9) | 77,69(2) | 76,78(3) | 48,7(17) | 56,08(15) | 51,43(16) | 71,71(8) | 73,42(4) | 73,42(4) | 67,02±3,77(10) |
| cardio | **95,55(1)** | 93,91(5) | 66,33(16) | 89,93(9) | 71,41(15) | 84,67(11) | 76,64(12) | 73,25(14) | 92,35(7) | 94,44(4) | 58,96(17) | 75,01(13) | 90,34(8) | 93,19(6) | 95,47(2) | 95,08(3) | 89,59±16,24(10) |
| Cardiotocography | 74,67(5) | 77,86(2) | 59,51(13) | 64,54(10) | 53,77(15) | 60,86(12) | 56,23(14) | 51,69(17) | 67,02(9) | 68,92(7) | 53,53(16) | 62,01(11) | 73,65(6) | 67,57(8) | 75,07(4) | 75,22(3) | **78,13±6,76(1)** |
| celeba | **79,38(1)** | 70,7(8) | 38,55(17) | 73,99(6) | 38,58(16) | 76,18(4) | 59,63(12) | 47,85(14) | 75,68(5) | 72,82(7) | 50,36(13) | 44,74(15) | 60,11(11) | 70,41(9) | 78,36(3) | 78,37(2) | 67,51±5,36(10) |
| census | 68,74(2) | 54,58(11) | 47,19(13) | 59,41(9) | 41,35(14) | 64,94(5) | 66,75(4) | 62,31(6) | **69,07(1)** | 68,44(3) | 51,07(12) | 59,29(10) | 36,86(15) | 59,52(8) | 0(16) | 0(16) | 61,16±5,21(7) |
| cover | **93,73(1)** | 92,62(5) | 84,58(13) | 89,3(8) | 76,91(15) | 80,24(14) | 85,97(12) | 74,46(16) | 88,64(10) | 93,42(3) | 46,2(17) | 89,89(7) | 92,34(6) | 86,74(11) | 93,43(2) | 93,42(3) | 88,72±16,18(9) |
| fault | 46,02(13) | 47,69(12) | 58,93(6) | 64,06(4) | 62,1(5) | 51,28(9) | 72,98(2) | 68,11(3) | 43,88(15) | 43,41(16) | 51,67(8) | 45,86(14) | 41,71(17) | 57,02(7) | 48,38(11) | 48,45(10) | **76,28±11,09(1)** |
| fraud | 90,35(11) | 90,62(9) | 94,92(2) | 91,7(8) | 93,05(7) | 90,29(12) | 93,56(6) | **94,97(1)** | 88,32(16) | 89,85(13) | 64,98(17) | 89,53(14) | 88,99(15) | 90,38(10) | 94,45(4) | 94,48(3) | 94,54±17,99(3) |
| glass | 66,29(15) | 35,36(17) | 69,2(14) | 82,94(2) | 72,24(12) | 77,23(4) | 82,29(3) | 73,36(8) | 72,43(11) | 75,7(7) | 47,49(16) | 76,09(6) | 73,13(9) | 77,13(5) | 73,11(10) | 71,25(13) | **87,1±3,47(1)** |
| Hepatitis | 75,95(5) | 67,75(10) | 38,02(17) | 66,4(11) | 41,45(16) | 79,85(5) | 52,76(14) | 68,17(9) | 82,05(2) | 79,67(4) | 50,96(15) | 54,8(13) | 64,87(12) | 69,75(8) | 74,88(6) | 73,54(7) | **83,37±5,29(1)** |
| http | 99,72(2) | 99,59(5) | 27,46(14) | 99,6(3) | 88,78(11) | 99,53(7) | 3,37(16) | 78,04(12) | 99,29(9) | 98,1(10) | 69,05(13) | - | 12,48(15) | **99,96(1)** | 99,6(3) | 99,59(5) | 99,48±47,51(8) |
| InternetAds | 61,67(12) | 68,28(4) | 65,83(8) | **70,58(1)** | 63,79(10) | 68,03(5) | 69,99(2) | 61,85(11) | 67,1(6) | 67,1(6) | 60,2(15) | - | 55,38(16) | 69,01(3) | 60,97(13) | 60,97(13) | 65,72±8,9(9) |
| Ionosphere | 79,19(11) | 75,92(13) | 90,59(2) | **90,72(1)** | 86,76(4) | 62,49(16) | 88,26(3) | 86,41(5) | 79,34(10) | 75,59(14) | 50,89(17) | 73,41(15) | 78,42(12) | 84,5(6) | 80,78(8) | 79,39(9) | 80,88±9,68(7) |
| landsat | 35,76(17) | 36,15(16) | 53,9(8) | 63,55(3) | 53,5(9) | 55,14(7) | 57,95(5) | 59,54(4) | 41,55(13) | 56,61(6) | 63,61(2) | 43,92(12) | 38,17(14) | 47,64(11) | 37,49(15) | 50,88(10) | **65,21±7,46(1)** |
| letter | 50,29(15) | 46,18(17) | 84,49(2) | 75,62(6) | 80,03(5) | 59,74(8) | **86,19(1)** | 84,09(3) | 54,32(10) | 50,76(13) | 56,64(9) | 50,42(14) | 50,24(16) | 61,07(7) | 51,58(12) | 52,68(11) | 82,34±8,02(4) |
| Lymphography | 99,82(5) | 99,54(7) | 89,86(12) | 99,83(4) | 90,85(11) | 99,49(9) | 55,91(16) | 72,49(14) | 99,48(10) | 99,52(8) | 32,29(17) | 72,11(15) | 85,55(13) | 99,81(6) | 99,84(2) | 99,84(2) | **99,98±11,89(1)** |
| magic.gamma | 67,22(10) | 60,65(15) | 68,51(7) | 75,13(3) | 66,64(12) | 70,86(6) | **82,38(1)** | 75,4(2) | 68,33(8) | 64,36(14) | 60,26(16) | 58,58(17) | 68,02(9) | 73,25(5) | 66,53(13) | 66,86(11) | 73,96±3,7(4) |
| mammography | 88,72(4) | 84,95(7) | 74,39(15) | 83,74(10) | 77,53(14) | 86,27(6) | 84,53(8) | 81,51(13) | 90,69(2) | **90,75(1)** | 56,98(16) | - | 83,91(9) | 86,39(5) | 83,55(11) | 88,86(3) | 82,07±29,37(12) |
| mnist | 85,29(2) | 82,95(6) | 67,13(14) | 79,45(9) | 70,78(12) | 60,42(15) | 80,58(8) | 60,1(16) | 77,74(10) | 84,6(5) | 53,4(17) | 67,23(13) | 72,27(11) | 80,98(7) | **85,44(1)** | 84,99(3) | 84,67±8,61(4) |
| musk | 100(1) | 80,58(11) | 41,18(16) | 100(1) | 38,69(17) | 100(1) | 100(1) | 69,89(14) | 74,09(13) | 94,2(10) | 95,11(8) | 43,52(15) | 76,85(12) | 95,11(8) | 99,99(7) | 100(1) | 100(1) |
| optdigits | 51,65(12) | 54(11) | 56,1(10) | 87,51(2) | 49,15(15) | 81,63(3) | 41,73(16) | 58,92(9) | 68,71(5) | 61,04(8) | 38,89(17) | 62,57(6) | 61,74(7) | 70,92(4) | 50,71(14) | 51,25(13) | **97,76±26,28(1)** |
| PageBlocks | 90,64(4) | 88,76(8) | 75,9(15) | 85,04(10) | 72,65(16) | 80,58(13) | 81,94(12) | 77,75(14) | 88,05(9) | 90,92(2) | 57,77(17) | 89,61(6) | 83,34(11) | 89,57(7) | **92,08(1)** | 90,66(3) | 90,5±4,42(5) |
| pendigits | 93,73(4) | 93,75(3) | 47,99(15) | 90,4(10) | 45,07(16) | 93,04(7) | 72,95(12) | 66,29(13) | 90,68(9) | 91,22(8) | 39,92(17) | 64,22(14) | 89,1(11) | 94,76(2) | 93,72(5) | 93,71(6) | **97,41±23,13(1)** |
| Pima | 70,77(5) | 66,92(8) | 65,71(10) | 71,42(3) | 61,05(13) | 71,07(4) | **73,43(1)** | 61,25(12) | 69,1(6) | 51,54(16) | 51,03(17) | 55,92(15) | 65,93(9) | 72,87(2) | 59,94(14) | 64,18(11) | 68,98±3,18(7) |
| satellite | 59,62(13) | 59,02(14) | 55,88(15) | 71,32(3) | 54,74(17) | 74,8(2) | 65,18(7) | 63,96(8) | 63,2(9) | **75,06(1)** | 55,3(16) | 62,33(10) | 61,98(11) | 70,43(4) | 61(12) | 69,61(5) | 69,15±6,95(6) |
| satimage-2 | 97,62(7) | 97,35(9) | 47,36(17) | **99,84(1)** | 56,7(15) | 97,65(6) | 92,6(13) | 83,08(14) | 97,21(10) | 97,11(11) | 53,14(16) | 96,29(12) | 97,56(8) | 99,16(3) | 97,68(5) | 98,59(4) | 99,79±37,18(2) |
| shuttle | 98,62(8) | 97,4(10) | 57,11(15) | 83,48(11) | 51,72(17) | 98,63(7) | 69,64(12) | 69,51(13) | 99,35(3) | 99,4(2) | 52,05(16) | 97,92(9) | 60,95(14) | **99,56(1)** | 99,01(5) | 98,98(6) | 99,13±26,94(4) |
| smtp | 88,41(4) | 80,7(7) | 71,84(13) | 79,68(8) | 79,6(9) | 70,52(15) | 89,62(3) | 59,85(17) | 79,09(10) | 71,86(12) | 78,24(11) | 71,32(14) | 67,43(16) | 89,73(2) | 82,23(5) | 81,66(6) | **96,77±8,46(1)** |
| SpamBase | 54,66(9) | 52,47(12) | 43,33(14) | 54,97(7) | 40,96(16) | 64,74(5) | **99,5(1)** | 91,72(14) | 91,9(13) | 99,42(2) | 97,2(10) | 65,69(17) | 76,67(16) | 98,26(8) | 98,95(5) | 98,72(7) | 70,84±17,04(1) |
| speech | 50,79(10) | 50,19(14) | 52,48(7) | 50,58(13) | **55,97(1)** | 50,59(12) | 51,03(9) | 55,86(2) | 52,89(5) | 51,58(8) | 53,43(4) | 52,75(6) | 49,84(15) | 50,74(11) | 46,89(16) | 46,89(16) | 55,53±2,93(3) |
| Stamps | 91,47(5) | 83,86(11) | 51,26(17) | 68,18(14) | 53,81(16) | 90,73(8) | 68,61(13) | 73,26(12) | 93,4(2) | 91,41(6) | 55,84(15) | 88,88(9) | 87,18(10) | 91,21(7) | 92,65(3) | 92,48(4) | **94,56±18,47(1)** |
| thyroid | 96,34(5) | 87,92(13) | 86,86(14) | 94,73(9) | 90,87(12) | 95,62(7) | 95,93(6) | 92,81(11) | 94,3(10) | 97,78(2) | 49,64(17) | 79,75(15) | 74,3(16) | **98,3(1)** | 96,74(4) | 95,55(8) | 97,08±9,13(3) |
| vertebral | 37,06(11) | 37,99(8) | 49,29(3) | 41,41(6) | 48,71(4) | 28,56(16) | 33,79(14) | 40,32(7) | 25,64(17) | 37,51(10) | 36,67(12) | 53,2(2) | 30,57(15) | 35,66(13) | 45,26(5) | 37,58(9) | **71,15±2,91(1)** |
| vowels | 65,29(9) | 61,59(11) | 93,12(3) | 89,92(5) | 94,04(2) | 72,21(7) | **97,26(1)** | 92,65(4) | 53,15(14) | 45,81(17) | 52,49(15) | 60,58(12) | 70,36(8) | 73,94(6) | 59,25(13) | 61,91(10) | 47,43±13,94(16) |
| Waveform | 65,48(11) | 56,29(15) | 73,32(3) | 72,42(7) | 72,56(5) | 68,77(9) | 73,78(2) | 68,57(10) | **75,03(1)** | 73,25(4) | 54,47(16) | 49,35(17) | 60,13(14) | 71,47(8) | 63,9(12) | 63,86(13) | 72,43±6,34(6) |
| WBC | 98,2(10) | 99,03(7) | 54,17(16) | 99,46(3) | 60,9(14) | 98,72(9) | 90,56(13) | 94,6(12) | 99,11(5) | 99,11(5) | 55,5(15) | - | 96,91(11) | 99,01(8) | 99,28(4) | 99,57(2) | **99,66±0,53(1)** |
| WDBC | 99,05(4) | 98,86(6) | 89(15) | 99,32(3) | 96,26(12) | **99,5(1)** | 91,72(14) | 91,9(13) | 99,42(2) | 97,2(10) | 65,69(17) | 76,67(16) | 98,26(8) | 98,95(5) | 98,72(7) | 98,72(7) | 55,23(16) |
| Wilt | 20,39(17) | 31,28(15) | 50,65(2) | 32,54(13) | 49,66(3) | 32,49(14) | 48,42(4) | **53,25(1)** | 33,4(11) | 39,43(8) | 46,08(6) | 37,29(9) | 26,42(16) | 41,94(7) | 34,51(10) | 33,03(12) | 46,66±13,48(5) |
| wine | 84,37(4) | 73,07(9) | 37,74(16) | 25,86(17) | 44,44(15) | **91,36(1)** | 44,98(14) | 46,11(13) | 88,65(3) | 71,34(10) | 59,52(12) | 61,7(11) | 90,12(2) | 80,37(6) | 73,1(8) | 78,23(7) | 83,79±3,12(5) |
| WPBC | 46,01(13) | 45,35(15) | 41,41(17) | 44,77(16) | 45,88(14) | **51,24(1)** | 46,59(12) | 51,14(2) | 49,34(7) | 46,83(10) | 49,79(3) | 47,8(9) | 49,31(8) | 46,63(11) | 49,6(4) | 49,53(5) | 49,4±7,51(6) |
| CIFAR10 | 63,87(9) | 63,76(10) | **68,57(1)** | 64,23(6) | 64,7(5) | 57,5(16) | 64,75(4) | 64,22(7) | 58,64(14) | 61,04(13) | 56,04(17) | 58,08(15) | 62,34(11) | 61,28(12) | 65,978(3) | 65,735(5) | 64,133±8,71(8) |
| FashionMNIST | 86,09(4) | 85,24(7) | 67,75(18) | 88,17(2) | 71,44(14) | 78,68(13) | 86,6(3) | 81,73(10) | 81,07(11) | 83,63(9) | 67,29(16) | 80,28(12) | 84,89(8) | 85,35(6) | 85,735(5) | **89,557±8,18(1)** |
| MNIST-C | 73,75(8) | 72,21(11) | 68,27(15) | 80,86(3) | 69,81(14) | 70,82(13) | 81,26(2) | 74(7) | 71,26(12) | 72,64(10) | 51,85(17) | 58,56(16) | 74,37(5) | 73,74(9) | 74,04(6) | 74,62(4) | **82,38±13,54(1)** |
| MVTec-AD | 72,42(9) | 69,84(13) | 74,19(3) | 75,98(1) | 69,7(14) | 73,36(5) | 72,96(7) | 71,57(10) | 72,91(8) | 73,46(4) | 57,1(17) | 66,47(16) | 68,51(15) | 73,19(6) | 70,872(12) | 71,248(11) | 75,49±13,25(2) |
| SVHN | 60,53(7) | 60,73(5) | **64,51(1)** | 60,3(8) | 63,47(2) | 56,08(16) | 62,63(3) | 61,09(4) | 56,75(15) | 58,27(12) | 53,47(17) | 57,22(14) | 58,26(13) | 58,62(11) | 60,00(9) | 60,697(6) | 59,531±4,85(10) |
| Agnews | 54,7(11) | 54,34(12) | **71,8(1)** | 60,02(6) | 68,97(2) | 53,87(13) | 64,11(4) | 62,81(5) | 52,98(15) | 53,04(14) | 42,51(17) | 52,02(16) | 55,47(10) | 56,74(8) | 56,575(9) | 57,495(7) | 64,15±4,40(3) |
| Amazon | 55,06(11) | 54,14(15) | 56,11(10) | 57,36(4) | 56,96(5) | 56,52(8) | 60,03(2) | **60,05(1)** | 56,94(6) | 56,79(7) | 39,08(17) | 53,58(16) | 54,2(14) | 56,3(9) | 54,95(12) | 54,53(13) | 57,43±6,97(3) |
| Imdb | 47,06(15) | 46,07(17) | 48,71(10) | 49,35(7) | 49,64(6) | 49,1(8) | 47,83(11) | 49,86(5) | 50,68(4) | 50,73(2) | 47,67(14) | 46,43(16) | 49,09(9) | 47,81(12) | 47,72(13) | **51,79±7,83(1)** |
| Yelp | 60,71(12) | 60,28(13) | 67,09(4) | 64,9(6) | 66,11(5) | 61,85(10) | **69,84(1)** | 67,74(3) | 62,36(8) | 62,15(9) | 54,62(17) | 56,28(16) | 61,36(11) | 62,53(7) | 59,23(15) | 59,79(14) | 68,77±11,73(2) |
| 20news | 56,66(8) | 56,45(9) | 62,14(2) | 57,59(6) | 61,8(3) | 56,28(10) | 59,33(4) | 58,56(5) | 55,79(12) | 56(11) | 50,24(17) | 54,17(16) | 55,53(13) | 56,9(7) | 54,23(15) | 54,38(14) | **63,78±7,45(1)** |

Table 3: Experimental datasets descriptions.

| Data | Samples | Features | Anomaly | % Anomaly | Category |
|---|---|---|---|---|---|
| cover | 286048 | 10 | 2747 | 0,96 | Botany |
| speech | 3686 | 400 | 61 | 1,65 | Linguistics |
| pendigits | 6870 | 16 | 156 | 2,27 | Image |
| optdigits | 5216 | 64 | 150 | 2,88 | Image |
| Waveform | 3443 | 21 | 100 | 2,9 | Physics |
| musk | 3062 | 166 | 97 | 3,17 | Chemistry |
| Lymphography | 148 | 18 | 6 | 4,05 | Healthcare |
| WBC | 223 | 9 | 10 | 4,48 | Healthcare |
| FashionMNIST | 6315 | 512 | 315 | 5 | Image |
| MNIST-C | 10000 | 512 | 500 | 5 | Image |
| Imdb | 10000 | 768 | 500 | 5 | NLP |
| donors | 619326 | 10 | 36710 | 5,93 | Sociology |
| mnist | 7603 | 100 | 700 | 9,21 | Image |
| PageBlocks | 5393 | 10 | 510 | 9,46 | Document |
| skin | 245057 | 3 | 50859 | 20,75 | Image |
| Cardiotocography | 2114 | 21 | 466 | 22,04 | Healthcare |
| fault | 1941 | 27 | 673 | 34,67 | Physical |
| satimage-2 | 5803 | 36 | 71 | 1,22 | Astronautics |
| SpamBase | 4207 | 57 | 1679 | 39,91 | Document |
| MVTec-AD | - | - | - | - | Image |
| Agnews | 10000 | 768 | 500 | 5 | NLP |
| Amazon | 10000 | 768 | 500 | 5 | NLP |
| Yelp | 10000 | 768 | 500 | 5 | NLP |
| shuttle | 49097 | 9 | 3511 | 7,15 | Astronautics |
| Stamps | 340 | 9 | 31 | 9,12 | Document |
| landsat | 6435 | 36 | 1333 | 20,71 | Astronautics |
| breastw | 683 | 9 | 239 | 34,99 | Healthcare |
| letter | 1600 | 32 | 100 | 6,25 | Image |
| wine | 129 | 13 | 10 | 7,75 | Chemistry |
| WPBC | 198 | 33 | 47 | 23,74 | Healthcare |
| satellite | 6435 | 36 | 2036 | 31,64 | Astronautics |
| magic.gamma | 19020 | 10 | 6688 | 35,16 | Physical |
| thyroid | 3772 | 6 | 93 | 2,47 | Healthcare |
| Hepatitis | 80 | 19 | 13 | 16,25 | Healthcare |
| annthyroid | 7200 | 6 | 534 | 7,42 | Healthcare |
| Pima | 768 | 8 | 268 | 34,9 | Healthcare |
| http | 567498 | 3 | 2211 | 0,39 | Web |
| celeba | 202599 | 39 | 4547 | 2,24 | Image |
| glass | 214 | 7 | 9 | 4,21 | Forensic |
| CIFAR10 | 5263 | 512 | 263 | 5 | Image |
| cardio | 1831 | 21 | 176 | 9,61 | Healthcare |
| campaign | 41188 | 62 | 4640 | 11,27 | Finance |
| vertebral | 240 | 6 | 30 | 12,5 | Biology |
| InternetAds | 1966 | 1555 | 368 | 18,72 | Image |
| fraud | 284807 | 29 | 492 | 0,17 | Finance |
| SVHN | 5208 | 512 | 260 | 5 | Image |
| mammography | 11183 | 6 | 260 | 2,32 | Healthcare |
| Wilt | 4819 | 5 | 257 | 5,33 | Botany |
| census | 299285 | 500 | 18568 | 6,2 | Sociology |
| Ionosphere | 351 | 33 | 126 | 35,9 | Oryctognosy |
| 20newsgroups | - | - | - | - | NLP |
| WDBC | 367 | 30 | 10 | 2,72 | Healthcare |
| smtp | 95156 | 3 | 30 | 0,03 | Web |
| vowels | 1456 | 12 | 50 | 3,43 | Linguistics |
| backdoor | 95329 | 196 | 2329 | 2,44 | Network |
| ALOI | 49534 | 27 | 1508 | 3,04 | Image |
| yeast | 1484 | 8 | 507 | 34,16 | Biology |

