# OpenReview forum: "AnoRand - Deep Learning-Based Semi-Supervised Anomaly Detection with Synthetic Labels"
_ICLR.cc/2024/Conference — ICLR 2024 Conference Withdrawn Submission_

### Official Review · Reviewer_Upnj · 2023-10-16

**Soundness:** 2 fair
**Presentation:** 1 poor
**Contribution:** 2 fair
**Rating:** 3
**Confidence:** 5

**Summary:**

The authors introduce an approach named AnoRand, which employs a two-step approach integrating a noise detection block and an autoencoder block. Through the introduction of synthetic anomalies, the methodology harnesses both supervised binary cross-entropy objectives and the autoencoder reconstruction loss.

**Strengths:**

The authors delve into the critical problem of unsupervised anomaly detection, a crucial challenge in machine learning.

**Weaknesses:**

1. The manuscript's overall quality suffers from poor writing and organization. The proposed architecture, a fundamental element of the paper, lacks a comprehensive and coherent discussion of its rationale. This leaves the reader somewhat confused in understanding its significance. Furthermore, the section concerning synthetic label generation is unclear and would greatly benefit from improved clarity. Additionally, the text contains numerous typos errors and grammatical mistakes, which disrupt the reader's engagement with the material. The experimental section, while integral to substantiating the proposed approach, is marred by confusion and inconsistencies. Notably, there is a discrepancy regarding the allocation of datasets into train/test splits, with conflicting statements in Sections 5 and 5.1. This inconsistency affects the interpretation of results, particularly those presented in Figure 2a. It is imperative that the dataset under evaluation be explicitly specified for clarity and reproducibility.

2. The overall efficacy of the proposed approach remains in question, as the results presented do not decisively establish its effectiveness. Section 6, intended for result analysis, suffers from similar clarity issues and fails to sufficiently illuminate the compared baselines, dataset specifics, and implementation particulars. Without these critical details, it is challenging to discern the full scope and implications of the presented findings.

3. The methodological analysis lacks depth and rigor. For instance, the sensitivity analysis of the parameter $\omega$ reveals a concerning sensitivity to different parameter values. This sensitivity raises questions about the proposed approach's robustness and generalizability. A more thorough exploration of such sensitivities is essential to ensure the method's reliability.

In its current state, the manuscript falls short of acceptance standards. Significant revision is essential to address these shortcomings.

**Questions:**

See weaknesses.

---

### Official Review · Reviewer_T2ce · 2023-10-26

**Soundness:** 2 fair
**Presentation:** 2 fair
**Contribution:** 1 poor
**Rating:** 3
**Confidence:** 5

**Summary:**

This paper focuses on the task of anomaly detection and introduces a semi-supervised approach called AnoRand. AnoRand begins by creating pseudo-anomalies through the introduction of noise into the normal data samples. Subsequently, it concurrently optimizes a discriminator and an auto-encoder. The anomaly score is computed by combining the auto-encoder's reconstruction error with the binary classification outcomes.  Experiments on image datasets, NLP dataset, as well as healthcare datasets are presented.

**Strengths:**

(1) The logic flow of this paper is smooth and the paper is easy to understand.

(2) Extensive experiments on diverse datasets from different domains (vision, NLP, numerical data) are performed.

**Weaknesses:**

(1) This paper introduces an approach to anomaly detection by combining a discriminator and an auto-encoder. A well-known challenge in prior research is that reconstruction-based methods in anomaly detection can be vulnerable to the "shortcut" problem, where anomalies can still be reconstructed effectively. The paper lacks clarity in explaining how the proposed method mitigates this limitation.

(2) Some of the claims made in the paper appear overly assertive. For instance, the paper asserts that most anomaly detection methods rely on the assumption that anomalies are situated in low-density regions. However, this assumption might hold true primarily for density-estimation-based algorithms. Many other algorithms, such as reconstruction-based and embedding-based methods, do not depend on this assumption.

(3)  The selection of comparison methods in this paper appears somewhat outdated. It would be advisable to include more up-to-date anomaly detection techniques for comparison. This would enable a more comprehensive evaluation of the proposed algorithm's effectiveness against state-of-the-art methods.

(4) It would be beneficial to provide introductory descriptions of the comparison methods used in Figure 2. Including the full names of these methods before using acronyms would enhance the clarity of the figure.

(5) The manuscript lacks self-containment. Several crucial results, including those in Table 2 and Figure 9, are presented to the Appendix. The main manuscript is incomplete in this sense.

**Questions:**

Please refer to the weakness section for my questions. In addition, I am wondering if the code of the algorithm will be published?

---

### Official Review · Reviewer_3QvY · 2023-10-29

**Soundness:** 3 good
**Presentation:** 2 fair
**Contribution:** 1 poor
**Rating:** 3
**Confidence:** 4

**Summary:**

This work introduces an anomaly detection method that first utilizes a feed forward network to learn and generate pseudo anomalies, and then use them, together with normal training data, to train a binary discriminator for anomaly detection. The method is evaluated and shows effective performance on a large number of tabular datasets, including some image/text datasets that were converted to tabular data using pre-trained neural networks as feature extractor.

**Strengths:**

- The key idea of the proposed method is simple and plausible.
- The method is evaluated on a large number of diverse 57 datasets, and shows effective performance on many of the datasets.
- The method is clearly elaborated with good paper structure.

**Weaknesses:**

- There have been a number of very similar methods introduced in the literature, aiming to learn pseudo anomalies for enabling the training of a binary/one-class classifier for anomaly detection, see some of them in [1-4]. However, the work does not discuss these methods and what are the advantages of the proposed method compared to them. So, the novelty of the method is unclear.
- I appreciate the extensive efforts on obtaining empirical results on such a large set of datasets with 17 competing methods, but these methods are more baseline methods rather than the recent state-of-the-art. It also neglects the aforementioned closely related methods in the comparison.
- The method seems to be quite sensitive to the hyperparameter $w$
- Results of analyzing $\alpha$ in the anomaly scoring function are missing.
- The related work section, particularly the supervised and semi-supervised algorithms subsection, is badly written. It's suggested to discuss each group of related studies in a more informed way, rather than simply putting them together.


**References**
- [1] Adversarially Learned Anomaly Detection. https://arxiv.org/abs/1812.02288
- [2] One-Class Adversarial Nets for Fraud Detection. https://arxiv.org/abs/1803.01798
- [3] Perturbation Learning Based Anomaly Detection. https://arxiv.org/abs/2206.02704
- [4] SimpleNet: A Simple Network for Image Anomaly Detection and Localization. https://arxiv.org/abs/2303.15140

**Questions:**

Please refer to the comments above.